# Minimal transmission in an influenza A (H3N2) human challenge-transmission model within a controlled exposure environment

Jonathan S. Nguyen-Van-Tam[1], Ben Killingley[1¤a]*, Joanne Enstone[1], Michael Hewitt[1], Jovan Pantelic[2¤b], Michael L. Grantham[2¤c], P. Jacob Bueno de Mesquita[2], Robert Lambkin-Williams[3], Anthony Gilbert[3], Alexander Mann[3], John Forni[3¤d], Catherine J. Noakes[4], Min Z. Levine[5], LaShondra Berman[5], Stephen Lindstrom[5], Simon Cauchemez[6¤e], Werner Bischoff[7], Raymond Tellier[8], Donald K. Milton[2], for the EMIT Consortium[¶]

1 Health Protection and Influenza Research Group, Division of Epidemiology and Public Heath, University of Nottingham School of Medicine, Nottingham, United Kingdom, 2 University of Maryland School of Public Health, Maryland Institute for Applied Environmental Health, College Park, Maryland, United States of America, 3 hVIVO London, United Kingdom, 4 University of Leeds School of Civil Engineering, Leeds, United Kingdom, 5 Centers for Disease Control and Prevention, Influenza Division, Atlanta, Georgia, United States of America, 6 Imperial College London, MRC Centre for Outbreak Analysis and Modelling, Department of Infectious Disease Epidemiology, London, United Kingdom, 7 Wake Forest School of Medicine, Winston-Salem, North Carolina, United States of America, 8 McGill University, Dept of Medicine, Montreal, Canada

¤a Current address: University College London Hospital, London, United Kingdom
¤b Current address: University of California, Center for the Built Environment, Berkeley, California, United States of America
¤c Current address: Missouri Western State University, St. Joseph, MO, United States of America
¤d Current address: Department of Acute and Specialty Care, MSD, London, United Kingdom
¤e Current address: Mathematical Modelling of Infectious Diseases Unit, CNRS UMR2000: Génomique évolutive, modélisation et santé (GEMS), Center of Bioinformatics, Biostatistics and Integrative Biology, Institut Pasteur, Paris, France
¶ A complete list of the EMIT Consortium can be found in the S1 Text.
* ben.killingley@nhs.net

**Data Availability Statement:** The study data are available in the public repository at Nottingham University at: https://rdmc.nottingham.ac.uk/

## Abstract

Uncertainty about the importance of influenza transmission by airborne droplet nuclei generates controversy for infection control. Human challenge-transmission studies have been supported as the most promising approach to fill this knowledge gap. Healthy, seronegative volunteer 'Donors' (n = 52) were randomly selected for intranasal challenge with influenza A/Wisconsin/67/2005 (H3N2). 'Recipients' randomized to Intervention (IR, n = 40) or Control (CR, n = 35) groups were exposed to Donors for four days. IRs wore face shields and hand sanitized frequently to limit large droplet and contact transmission. One transmitted infection was confirmed by serology in a CR, yielding a secondary attack rate of 2.9% among CR, 0% in IR (p = 0.47 for group difference), and 1.3% overall, significantly less than 16% (p<0.001) expected based on a proof-of-concept study secondary attack rate and considering that there were twice as many Donors and days of exposure. The main difference between these studies was mechanical building ventilation in the follow-on study, suggesting a possible role for aerosols.

handle/internal/8311. The DOI is http://doi.org/10.17639/nott.7051.

**Funding:** This work was supported by U.S. CDC (https://www.cdc.gov/publichealthgateway/partnerships/index.html), Cooperative Agreement: Grant Number 1U01P000497-01. Employees of the funder participated in discussion of the study design, and sample analysis; these individuals also participated in manuscript preparation and are named as co-authors. This work was also supported by the National Institute of Allergy and Infectious Diseases Centers of Excellence for Influenza Research and Surveillance (CEIRS) and this funder had no role in study design, data collection and analysis, decision to publish, or preparation of the manuscript.

**Competing interests:** I have read the journal's policy and the authors of this manuscript have the following competing interests: JSN-V-T and BK declare previous consultancy fees from H-Vivo plc, unrelated to the current work. JSN-V-T is currently seconded to the Department of Health and Social Care (DHSC), England; the views expressed in this paper are not necessarily those of DHSC. RLW, AG and AM are employees of H-Vivo plc each of whom hold shares and /or share options in the company.

## Author summary

Understanding the relative importance of influenza modes of transmission informs strategic use of preventive measures to reduce influenza risk in high-risk settings such as hospitals and is important for pandemic preparedness. Given the increasing evidence from epidemiological modelling, exhaled viral aerosol, and aerobiological survival studies supporting a role for airborne transmission and the potential benefit of respirators (and other precautions designed to prevent inhalation of aerosols) versus surgical masks (mainly effective for reducing exposure to large droplets) to protect healthcare workers, more studies are needed to evaluate the extent of risk posed airborne versus contact and large droplet spray transmission modes. New human challenge-transmission studies should be carefully designed to overcome limitations encountered in the current study. The low secondary attack rate reported herein also suggests that the current challenge-transmission model may no longer be a more promising approach to resolving questions about transmission modes than community-based studies employing environmental monitoring and newer, state-of-the-art deep sequencing-based molecular epidemiological methods.

## Introduction

Influenza virus is a pathogen of global health significance, but human-to-human transmission remains poorly understood. In particular, the relative importance of the different modes of transmission (direct and indirect contact, large droplet, and aerosols (airborne droplet nuclei)) remains uncertain during symptomatic and asymptomatic infection [1–4].

The evidence base for influenza transmission is derived from studies that have assessed: virus deposition and survival in the environment; the epidemiology of disease; pharmaceutical and non-pharmaceutical interventions; animal models; and mathematical models of transmission. Those approaches have yet to produce conclusive data quantifying the relative importance of human-human transmission modes [1,2].

Infection control guidance for pandemic and seasonal influenza assumes that most transmission occurs during symptomatic infection, predominantly via large droplet spread at short range (1-2m) [1]. Thus, social distancing measures are often proposed to mitigate the spread and impact of a pandemic; and hand washing and respiratory etiquette are promoted to reduce transmission. Evidence to support the possibility of aerosol transmission has grown over recent years [5–7]. and leads to controversies about when and if filtering facepiece respirators (and other precautions designed to prevent inhalation of aerosols) versus surgical masks (mainly capable of reducing large droplets and some fine particles) should be used to protect healthcare workers, particularly during a severe pandemic [1,3,4,8–10].

An expert panel, after in-depth review of the challenges facing community- and workplace-based intervention studies and their failure thus far to provide definitive evidence regarding the relative contribution of the various modes, concluded that a human challenge-transmission study would be a more promising direction for future research [11]. Influenza challenge studies in humans have been conducted to investigate disease pathogenesis and the efficacy of antivirals and vaccines. Challenge studies assessing human-to-human transmission had not been performed [11]. In 2009, we demonstrated proof-of-concept that healthy seronegative volunteers inoculated intranasally with influenza A/Wisconsin/67/2005 (A/WI), an H3N2 virus, would develop symptoms of influenza-like illness (ILI) and, under two days of household-like conditions without environmental controls, transmit infection to other seronegative

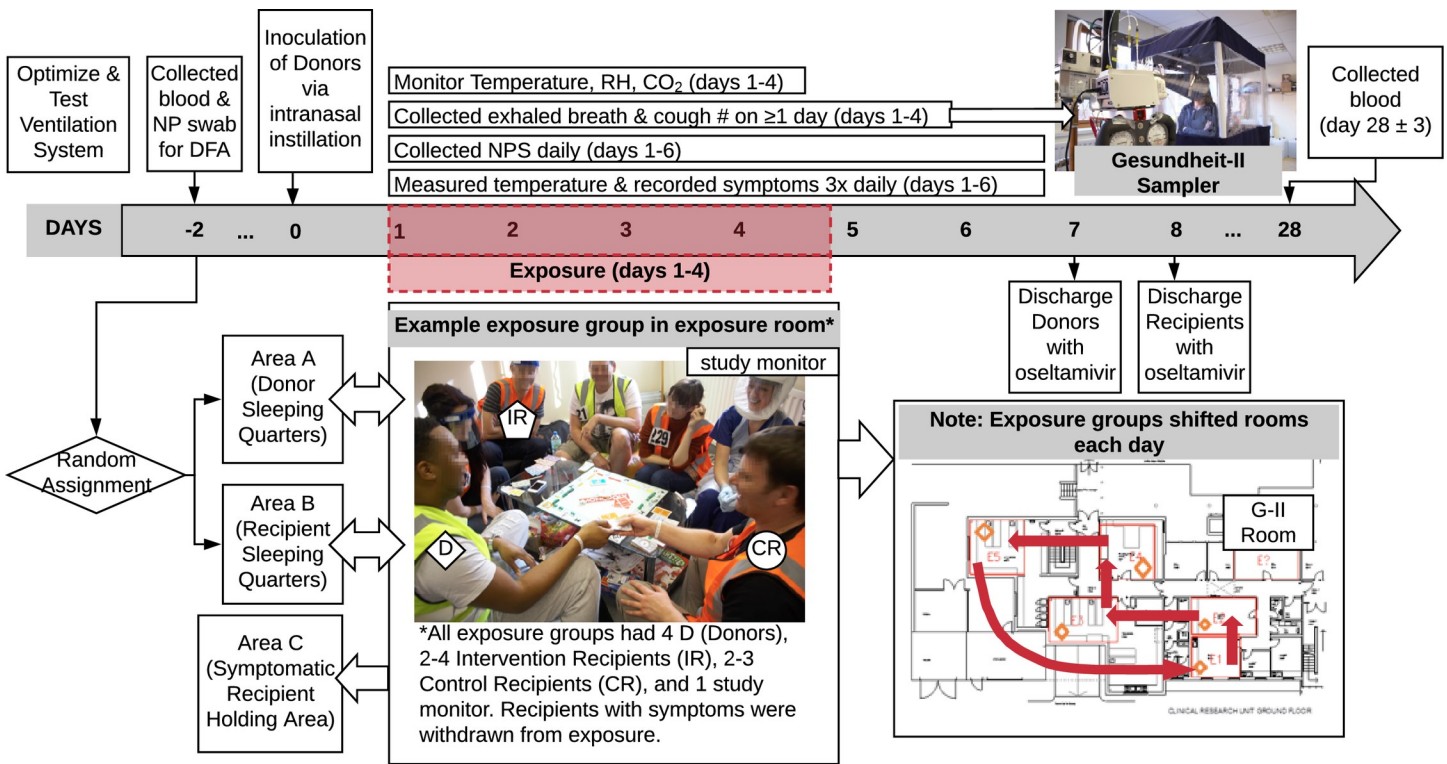

**Fig 1. Schematic of study design showing timelines, environmental controls and monitoring, physical segregation arrangements, exposure intervention, and volunteer movements during quarantine study.** DFA: direct fluorescence assay; RH: relative humidity; NPS: nasopharyngeal swab.

volunteers. This suggested that larger scale human challenge-transmission models might be useful to evaluate transmission modes. A subsequent international workshop discussed the potential that human challenge-transmission studies, with appropriate interventions, monitoring of aerosol shedding, and environmental controls, could provide definitive results [12]. Here, we report a large follow-on study, including design factors (Fig 1) aimed at assessing the importance of aerosol transmission in human-to-human transmission of influenza virus. Although the study did not achieve the intended level of transmission required for a more conclusive interpretation, it has revealed important lessons about potential airborne transmission and study design that represent critical knowledge to support effective large-scale, costly studies in this area.

## Results

### Participation and safety

Between January and June 2013, 496 seronegative (HAI≤10 to the challenge virus antigen) volunteers underwent study-specific screening and 166 entered the quarantine unit, of whom 127 proved suitable for final study entry (Fig 2 –trial profile; S5 Text–volunteer baseline characteristics). Thirty-nine subjects were discharged before inoculation or exposure per protocol as described in Methods. Three separate quarantine EEs (exposure events) took place in March, April, and June 2013 involving Q1: 41 (20 Donors; 11 CR; 10 IR), Q2: 31 (12 Donors; 9 CR; 10 IR) and Q3: 55 (20 Donors; 15 CR; 20 IR) subjects respectively, with 4 Donors and 4 to 7 Recipients per exposure group. No serious adverse events were recorded in volunteers who commenced the study.

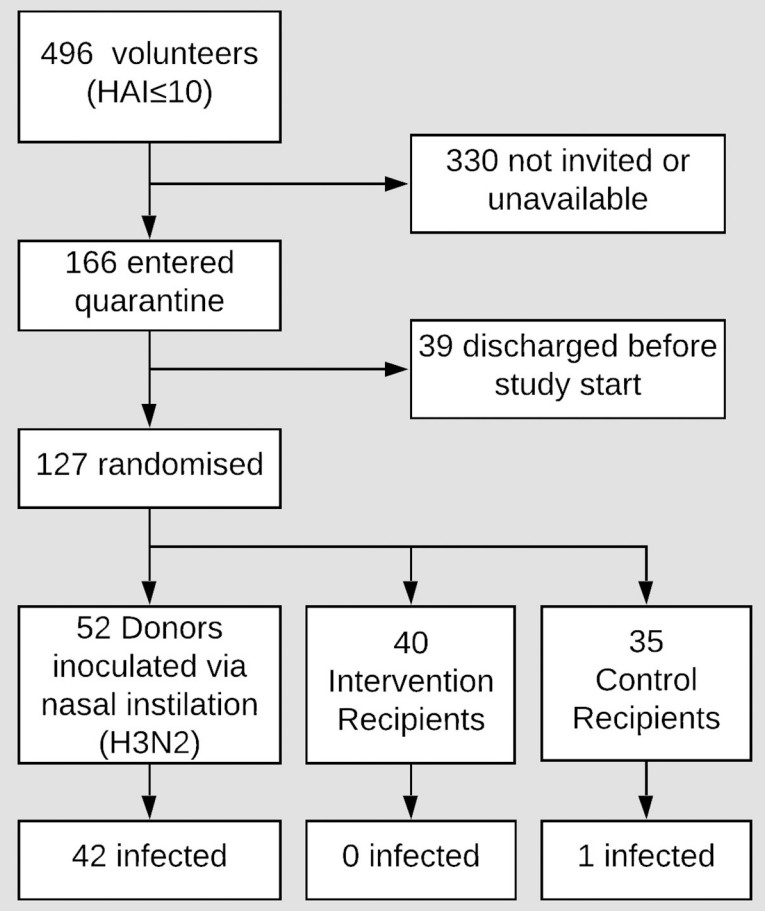

**Fig 2. Trial profile.** Intervention Recipients: wore face shields, used hand sanitizer every 15 min and only allowed to touch face with single-use wooden spatula; Control Recipients: did not use face shields or the specified hand hygiene protocol.

## Environmental control

In Q1 relative humidity averaged 40% (Standard Deviation 9%), room temperature averaged 20.2 °C (0.4 °C) and $CO_2$ concentration averaged 1430ppm (110ppm). For Q2 and Q3, respectively, the corresponding values were 44% (4%), 21.4°C (0.3 °C), 1810ppm (160ppm), and 57% (4%), 21.4°C (0.3 °C), 1810ppm (160ppm). Outdoor $CO_2$ concentration proxies, taken from the average of $CO_2$ measurements during 2:00am-3:00am were 418, 435, and 422 ppm, for Q1, Q2, and Q3, respectively.

## Donor status

Donor status is summarised by quarantine study in Table 1. Over all quarantines combined, intranasal inoculation produced an infection rate of 81% (42/52) among inoculated volunteers. Of the 42 lab-confirmed infected Donors 25 (60%) had ILI and 10 (24%) were classified as asymptomatic (4 in Q1, 4 in Q2, and 2 in Q3).

Ten Donors had greater immunity on admission, as identified by samples collected on day -2 (HAI>10 or MN≥80), than at their earlier screening. Four of the 10 seroconverted (i.e. had a 4-fold rise in HAI or MN titres) between admission to quarantine and follow-up. Five of the

**Table 1. Infected donor status.**

| Q* | Infected/Inoculated n/n (%) | Clinical Illness n (% of Infected) | | | Laboratory-confirmed Infection n (% of Infected) | | |
|---|---|---|---|---|---|---|---|
| | | Symptomatic | Febrile | ILI | PCR-confirmed | PCR-confirmed & Seroconversion | Seroconversion by HAI: MN: Either |
| 1 | 15/20 (75) | 11 (73) | 4 (27) | 8 (53) | 12 (80) | 11 (73) | 12: 14: 14 |
| 2 | 11/12 (92) | 7 (64) | 0 (0) | 5 (45) | 10 (91) | 8 (73) | 9: 7: 9 |
| 3 | 16/20 (80) | 14 (88) | 2 (13) | 12 (75) | 14 (88) | 12 (75) | 14: 11: 14 |
| All | 42/52 (81) | 32 (76) | 6 (14) | 25 (60) | 36 (86) | 31 (74) | 35: 32: 37 |

*Quarantine number; Ten Donors had greater than anticipated pre-challenge immunity: 4 in Q1, 2 in Q2, 4 in Q3.

10 met laboratory case definition by qRT-PCR including all four who seroconverted. The one additional qRT-PCR positive Donor had positive swabs on study days 2 and 3 in Q2.

## Virus shedding by donors

Overall, 36 Donors had nasopharyngeal swab (NPS) that tested positive by PCR for A/WI. Of these 36: 53% (n = 19) were positive on day 1 post-challenge; 94% (34) on day 2; 97% (35) on day 3; 86% (31) on day 4; 92% (33) on day 5; and 67% (24) on day 6 (Fig 3).

Aerosol shedding was determined for 25 Donors on day 1, 31 on day 2, 30 on day 3, and 24 on day 4, and for a total of 36 person-days in Q1, 34 person-days in Q2, and 40 person-days in Q3. Aerosol shedding from infected Donors, detected in Gesundheit-II samples, is summarised in Table 2. Six (7%) of the coarse and 14 (16%) of the fine aerosol samples had detectable viral RNA. We observed aerosol shedding from 11 (26%) of the 42 successfully infected Donors. The geometric mean (GM) and geometric standard deviation (GSD) for coarse and fine aerosol viral RNA copy numbers per 30-min sample were 3.1E+3 (3.3) and 5.3E+3 (4.6), respectively. The maximum levels of shedding into coarse and fine aerosols were 2.79E+4 and 8.02E+4 RNA copies, respectively (Fig 3).

## Recipient status

Recipient status is shown in Table 3. There were similar rates of symptoms in both IR and CR groups, although more in the CR group met the study definition of ILI, the rates were not significantly different (p = 0.23); no Recipient developed fever. One infection was confirmed by serology (HAI increased from ≤10 to 40 and MN increased from 10 to 320) in a CR subject who was symptomatic and whose symptoms met the definition of ILI, but whose qRT-PCR evaluations were persistently negative. Two other CR were transiently qRT-PCR positive but neither met laboratory positivity criteria (S6 Text). Both were asymptomatic and had no change from baseline serology. Thus, there was only one, confirmed transmission event. The CR and IR group SARs (2.9% and 0%) were not significantly different (p = 0.47).

To compare these results with the SAR from the proof-of-concept study, we recomputed the latter results using the current, more stringent outcome criteria. The adjusted proof-of-concept SAR was 8.3% giving an expected SAR of 16%. The observed SAR for the current study was not significantly different than that of the adjusted SAR from the proof-of-concept study, but was significantly lower than the expected doubling of the SAR (1.3% overall, p<0.0001; and 2.9% for CR, p = 0.035). Comparisons of observed and expected SAR using the proof-of-concept study outcome definitions were also statistically significant (p<0.0001, S6 Text).

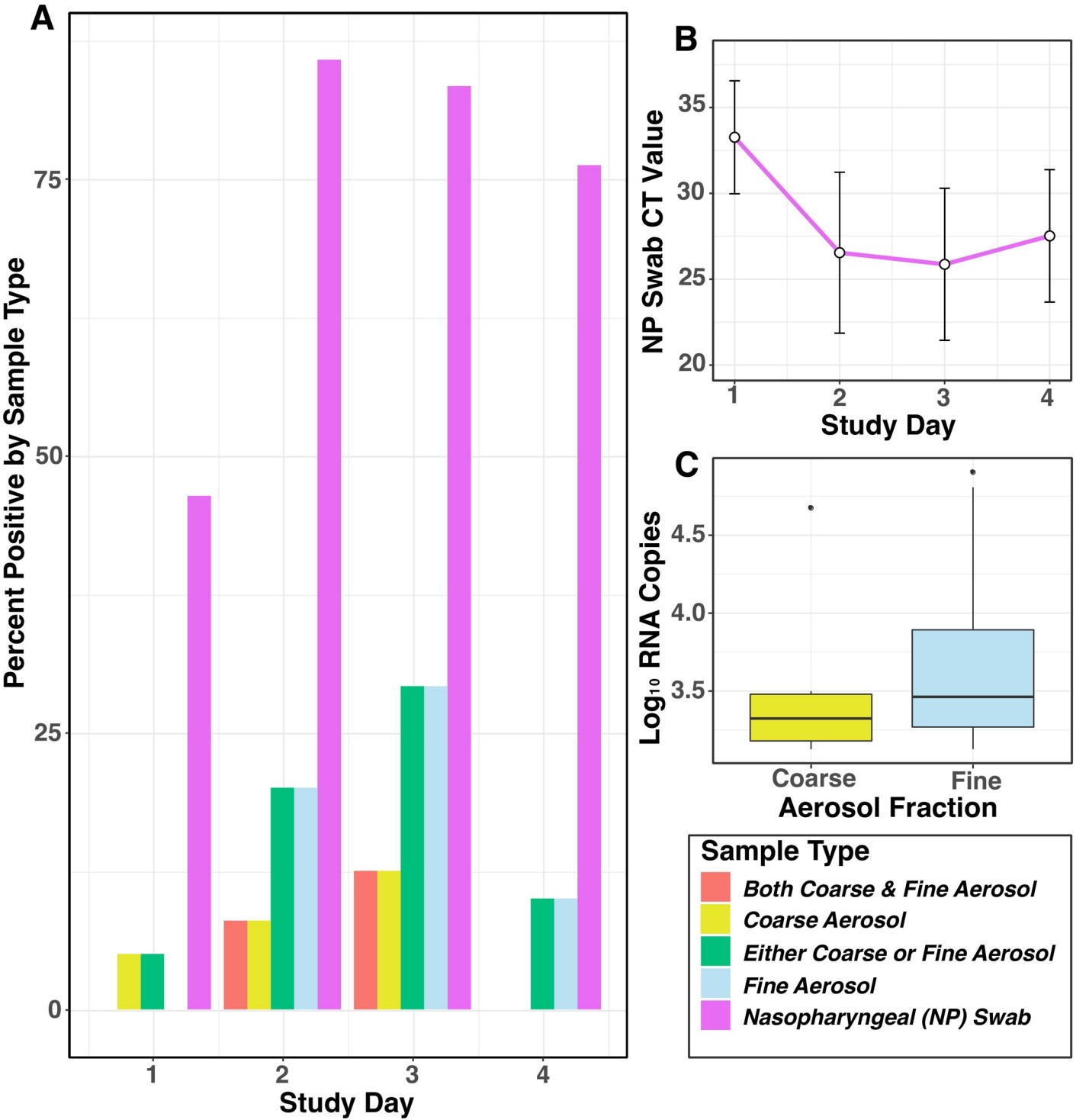

**Fig 3. Viral detection in Donors by day of exposure event.** A) Columns show the proportion of all infected donors (n = 42) who were qRT-PCR positive for viral shedding for coarse (>5μm) and fine (≤5μm) aerosols, and nasopharyngeal swabs. B) Mean and standard deviation error bars for qRT-PCR cycle threshold values from the positive nasopharyngeal swabs (n = 19 on day 1; n = 34 on day 2; n = 35 on day 3; n = 31 on day 4). C) Virus quantified ($\log_{10}$ RNA copies) from detectable exhaled coarse (n = 6) and fine (n = 14) breath aerosols by qRT-PCR; the boxes show the inner-quartile range (IQR) with a band to indicate the median, and whiskers extending to highest and lowest data points within 1.5 IQR.

**Table 2. Exhaled breath viral RNA detection and copy number among infected donors by quarantine event and aerosol fraction.**

| Q* | n Subjects | n Samples | Coarse Aerosol (>5μm) | | | Fine Aerosol (≤5μm) | | |
|---|---|---|---|---|---|---|---|---|
| | | | Positive Subjects (%) | Positive Samples (%) | Positive Sample Mean[†] | Positive Subjects (%) | Positive Samples (%) | Positive Sample Mean[†] |
| 1 | 15 | 27 | 1 (7) | 1 (4) | 2.79e+04 | 3 (20) | 5 (19) | 3.32e+04 |
| 2 | 11 | 30 | 3 (27) | 3 (10) | 2.16e+03 | 3 (27) | 4 (13) | 1.80e+04 |
| 3 | 16 | 32 | 2 (13) | 2 (6) | 1.73e+03 | 5 (31) | 5 (16) | 1.70e+03 |
| All | 42 | 89 | 6 (14) | 6 (7) | 6.31e+03 | 11 (26) | 14 (16) | 1.76e+04 |

*Quarantine number

[†]The arithmetic mean RNA copy number used positive samples only; The geometric means (GM) and geometric standard deviations (GSD) over all positive samples were 3.14E+3 (3.33) and 5.31E+3 (4.59) for coarse and fine aerosol samples, respectively.

## Discussion

To our knowledge, this is the largest human influenza challenge-transmission study undertaken to date. We applied measures to control and standardise environmental conditions and ventilation rates within and between exposure events, to emulate as far as possible indoor winter conditions when respiratory virus spread is maximal. We particularly sought to maintain low humidity conditions which have been associated with enhanced transmission [13] and increased virus viability [14], together with a low ventilation rate to maximize recipient exposure to airborne virus. The near absence of transmission to control Recipients suggests contact and large droplet spray did not contribute substantially to transmission under the conditions used in these EEs. The significantly lower than expected SAR in this study compared with the proof-of-concept study, which had much lower ventilation rates, suggests aerosols as an important mode of influenza virus transmission in this model. The overall low SAR suggests that Donors in this model were minimally contagious and prevents definitive assessment of the modes of transmission.

Having reported an SAR of 25% (3/12) in our earlier proof-of-concept study, we expected to observe an SAR of >25%, having doubled both the duration of the exposure and the number of Donors in each quarantine [15]. Indeed, the study was designed to examine an SAR of

**Table 3. Recipient status.**

| Q* | Recipient[†] | Infected/Exposed n/n (%) | Clinical Illness n (% of Exposed) | | | Laboratory-confirmed Infection n (% of Exposed) | | |
|---|---|---|---|---|---|---|---|---|
| | | | Symptomatic | Febrile | ILI | PCR-confirmed | PCR-confirmed & Seroconversion | Seroconversion by HAI: MN: Either |
| 1 | CR | 0/11 (0) | 4 (36) | 0 (0) | 3 (27) | 0 (0) | 0 (0) | 0: 0: 0 |
| | IR | 0/10 (0) | 2 (20) | 0 (0) | 1 (10) | 0 (0) | 0 (0) | 0: 0: 0 |
| 2 | CR | 1/9 (11) | 2 (22) | 0 (0) | 2 (22) | 0 (0) | 0 (0) | 1: 1: 1 |
| | IR | 0/10 (0) | 3 (30) | 0 (0) | 2 (20) | 0 (0) | 0 (0) | 0: 0: 0 |
| 3 | CR | 0/15 (0) | 6 (40) | 0 (0) | 4 (27) | 0 (0) | 0 (0) | 0: 0: 0 |
| | IR | 0/20 (0) | 6 (30) | 0 (0) | 2 (10) | 0 (0) | 0 (0) | 0: 0: 0 |
| All | CR | 1/35 (3) | 12 (34) | 0 (0) | 9 (26) | 0 (0) | 0 (0) | 1: 1: 1 |
| | IR | 0/40 (0) | 11 (28) | 0 (0) | 5 (13) | 0 (0) | 0 (0) | 0: 0: 0 |

*Quarantine number; Eleven recipients had greater than anticipated pre-challenge immunity: 3 CR and 3 IR for Q1; no Q2 Recipients; 3 CR and 2 IR for Q3. None seroconverted.

[†]CR: Control Recipient; IR: Intervention Recipient.

40% in CR versus 20% in IR which would have required 125 Recipient volunteers; this was not met (n = 75) and the study was underpowered.

However, the outcome criteria used in the proof-of-concept study, which included as positive a single NPS positive by qRT-PCR without seroconversion, were made more stringent in the present study by requiring two or more NPS positive by qRT-PCR in the absence of seroconversion (S6 Text). Applying the proof-of-concept criteria to the current study gives an SAR of 4% (3/75) overall, while applying the stricter criteria used in this study to the proof-of-concept study gives an SAR of 8.3% (1/12) rather than 25%. Given the lower than planned enrolment and the stricter outcome criteria, this study was doubly underpowered.

These observations raise two questions: 1) Why were SARs, using stringent criteria, low in both studies and what are the implications for future human challenge-transmission studies? 2) Why was the SAR significantly lower in the present study compared with the expected doubling of the rate observed for the proof-of-concept?

The low SAR in these studies suggests that, unless a much greater SAR can be achieved, type II error associated with underpowering will be a major obstacle to successful use of human challenge-transmission studies. Potential areas to consider addressing in order to raise the SARs in future studies include the virus used, the route of inoculation, susceptibility of the human volunteers, the rate of viral shedding into NPS and aerosols, and reducing ventilation of exposure rooms.

In the proof-of-concept (2009) and follow-on (2013) studies we used a GMP A/WI challenge virus manufactured by Baxter BioScience (Vienna, Austria). Both studies produced similar clinical and serological infection rates (typically 60–70% and 70–75%, respectively) after inoculation via nasal instillation, and similar spectrums of clinical illness severity in Donors. These rates were higher than reported by a previous study using lower doses of the same viral preparation [16] and consistent with rates reported in a review of 56 challenge studies [17]. The illnesses we observed were similar to the range seen in healthy adults in the community, from asymptomatic to febrile symptomatic infection [18]. Thus, skewed illness severity does not seem to explain the low SAR.

The virus preparation has been used in other human challenge studies with similar rates of infection via nasal instillation [19]. Using deep sequencing, Sobel Leonard and colleagues showed that a sample of the Baxter stock "was at least partially adapted" to the egg and/or tissue environments in which it was produced [19]. They also found that nasal instillation of the stock into human volunteers resulted in rapid purification selection, although a fixed variant in the HA gene remained. We have performed a BLAST search and identified the fixed variant (G70A/D8N) in deposited sequences of wild-type H3N2 viruses. This suggests that, on its own, the fixed HA variant is unlikely to have been a key alteration. These results suggest that it is unlikely that the virus stock was the primary cause of the low SARs. But, the impact of positive selection of the challenge virus for growth in the production environment, rather than for human transmissibility, remains a potential contributing factor to consider in choosing challenge viruses for future transmission studies.

The route of infection with influenza virus is known to matter in the setting of experimental infection, with aerosolized virus infectious at lower doses and more likely to result in 'typical influenza-like disease' (fever plus cough) than intranasal inoculation [20,21]. This anisotropic property [22] of influenza virus is not unique among respiratory viruses; e.g. it is exploited by the live, unattenuated adenovirus vaccine [23]. The implication for human challenge-transmission studies, however, may be that increased rates of lower respiratory tract infection via aerosol inoculation might be required to achieve sufficiently high rates of donors with fever, cough, and contagiousness to achieve a useful SAR.

In the current quarantine-based human challenge-transmission model, consistent with historical precedent, screening for susceptibility was undertaken primarily by HAI antibody screening, although it is recognised that screening by MN titre or other assays [24,25] could be an alternative or adjunctive approach. The exact correlates of immunity and severity using novel immunological assays have not been validated and selecting subjects based on these assays would have added substantial complexity and costs. Six Donors and five Recipients in the present study were discovered, in retrospect, to have seroconverted during the 3 to 56-day interval prior to entering the quarantine facility, despite having as short a delay as possible between final screening for HAI and quarantine entry. However, the majority of Donors and Recipients were susceptible according to the results of microneutralization tests. Prior immunity, as measured by the HAI and MN assays, does not therefore, appear to have been a major limitation nor account for failure to transmit from readily infected Donors to identically screened Recipients. Regarding future studies, as novel immune correlates of influenza protection and severity become established, additional approaches beyond HAI and MN assays could be employed for volunteer selection. This might enable selection of those likely to become infected, febrile, and have greater symptomatology including more frequent and greater levels of cough and runny nose. Unfortunately, such screening might also dramatically reduce the yield of suitable volunteers and substantially increase overall study costs.

Results from serial nasopharyngeal swabs in Donors indicate that over 80% were positive by qRT-PCR testing on one or more post-challenge days. Viral load detected by swabs was substantial with qRT-PCR Ct values in the mid 20s on days two and three (Fig 3). Thus, failure to shed virus into nasal secretions cannot explain the low SARs. Minimal transmission despite days of prolonged exposure to Donors shedding virus into nasal secretions provides strong evidence that contact and large droplet transmission are not important in this model.

The results from breath sampling with the Gesundheit-II device indicate that 26% (11/42) of infected Donors had virus detectable in exhaled air during the same period. By comparison, virus shedding into exhaled breath was detected in 84% (119 of 142) of influenza cases selected on the basis of having fever or a positive rapid test and sampled on one to three days post onset of symptoms, mostly recruited from young adults on a college campus [7]. When compared on a per-sample basis, infected Donors shed detectable virus less frequently than naturally infected college campus cases [7] in both coarse (7% and 40%, respectively) and fine aerosols (15% and 76%); all assays for both groups were performed in the same laboratory using the same methods. However, when the comparison was limited to positive aerosol samples from each study population, the average quantities of virus detected were similar (within 1 log), for the Donors as for the college community cases (GM coarse 3.1E+3 and 1.2E+4, GM fine 5.3E+3 and 3.8E+4, respectively). The maximum exhaled breath viral aerosol from the 11 Donors was two to four logs lower than from the college campus cases selected for having fever or a high viral load in the NPS (maximum coarse 2.8E+4 and 4.3E+8, and maximum fine 8.0E+4 and 4.4E+7, respectively) [7]. While this difference may merely represent the low probability of sampling from the tail of a log-normal distribution with only 11, as compared with 119 cases, it may be relevant to the low SAR in the challenge-transmission model if aerosols disseminated by more symptomatic individuals, such as the selected community cases, and rare supershedders account for most transmission. If aerosols are largely derived from the lower respiratory tract, as has been suggested by analysis of the college community cases, this would also suggest that future challenge-transmission studies should employ methods designed to increase the frequency of lower respiratory tract infection.

The proof-of-concept study was conducted in a hotel room with closed windows and thermal control provided only by a recirculating air conditioning unit. While the ventilation rate was not measured, it was likely to have been extremely low for the number of occupants, with

only small, intermittent, bathroom extract and natural building infiltration providing fresh air. The ventilation rate of 4 l/s/person during the main study was low compared to 10 l/s/person recommended in UK design standards [26] but was likely substantially greater than in the proof-of-concept study. This would have produced significantly higher viral aerosol concentrations during the proof-of-concept EEs, assuming similar generation rates from Donors in both experiments. Given that the Donors in the two studies were similar in other respects, differences in shedding rates seem unlikely. Therefore, the difference in SAR between this study and the expected SAR based on design changes and prior results are possibly due to differing ventilation conditions. The implication for future challenge-transmission studies, given that the ventilation rate in the current study was as low as possible with a single pass ventilation system, is that recirculating air conditioning systems similar to that in the proof-of-concept study should be employed to limit dilution ventilation and maximize exposure to aerosols. This will be especially important if Donors in future studies continue to represent the lower end of the aerosol shedding spectrum seen in naturally infected cases.

Achieving temperature and humidity to simulate winter conditions was challenging, particularly in Q3, conducted in June 2013 when the average external conditions were 16˚C and 64% relative humidity. It was necessary to strike a balance between volunteer comfort and conditions favourable to transmission, both of which were constrained by the capability of the mechanical systems in the building. However, the relative humidity in the current study overlapped with that during the proof-of-concept study, which ranged between 38 and 53%, and thus, eliminated humidity as a potential explanation for the difference in transmission rates.

Despite this study not having produced the planned SAR, it yields important findings. First, although fewer viral challenged subjects had virus-laden aerosols than seen in people with natural infections presenting with influenza-like symptoms, those volunteers who did produce viral aerosols did so at a rate similar to the average symptomatic naturally infected case. Second, given that a subset of the infected volunteers had moderate viral aerosol shedding in this model, observation of transmission via aerosols in quarantine studies may be strongly dependent on the dilution ventilation rate. Third, low risk of transmission to Control Recipients suggests that contact and large droplet spray transmission were not important modes of transmission in this model. The overall low SAR compared to that observed in the proof-of-concept study suggests that, given the main difference between the studies was the indoor air ventilation rate, aerosol transmission may be an important mode of influenza virus transmission between adults. Finally, sensitivity of transmission to details of the Donors selected, environment, and activity during exposure events, suggest that if a successful transmission model can be developed, carefully designed studies may be useful for investigating specific, targeted intervention strategies for prevention of specific transmission modalities. However, sensitivity to experimental conditions also demonstrates that it will be challenging to generalize the results of the quarantine-based transmission model to broad conclusions about the relative importance of aerosol, droplet spray, and contact modes of transmission. These complexities of the challenge-transmission model suggest that community-based transmission studies employing deep-sequencing based molecular epidemiologic methods in natural experiments, e.g. comparing high and low ventilation dormitories or barracks, may be more attractive alternatives than previously thought. Unfortunately, although an important role for aerosols in transmission of influenza, at least between adults, is hinted at when comparing the proof-of-concept and current studies, this challenge model cannot provide a definitive answer to the importance of this mode for influenza virus transmission between humans.

## Materials and methods

### Ethics statement

The randomized challenge-transmission trial took place from March to June 2013 in a closed quarantine facility, with written informed consent from healthy volunteers in accordance with the principles of the Declaration of Helsinki, in compliance with UK regulatory and ethical (IRB) requirements (under auspices of the UK Health Research Authority (HRA) National Research Ethics Service (NRES) Committee London–City & East; reference number 12/LO/ 1277), and registered with ClinicalTrials.gov (number NCT01710111). The role of the funding source is described in S2 Text.

### Overview

Volunteers, screened for serologic susceptibility, were randomly selected for intranasal challenge with A/WI [15]–becoming 'Donors'. After allowing for a short incubation period, Donors were introduced to other sero-susceptible volunteers–'Recipients'–under controlled household-like conditions for four days. Recipients were randomised as Intervention Recipients (IR) or Control Recipients (CR). IRs wore face shields evaluated to interrupt large droplet transmission but to be permissive to aerosols (S3 Text); in addition, IRs hand sanitised (using alcohol-based Deb InstantFOAM, 72% ethyl alcohol) once every 15 minutes to minimise the possibility of contact transmission. IRs were only allowed to touch face via single-use wooden spatulas. Thus, IRs would be exposed to influenza only via aerosols. CRs did not wear face shields or use hand sanitiser and were allowed to touch face freely; therefore, CRs would have been exposed via all routes of transmission consistent with close proximity human-human contact. Fig 1 shows an overview of the study design.

### Influenza virus

Influenza A/WI manufactured and processed under current good manufacturing practices (cGMP) was obtained from Baxter BioScience, (Vienna, Austria). Stocks of this virus preparation have been sequenced and its evolution in the upper respiratory track of inoculated volunteers extensively analysed [19].

### Screening

Volunteers were screened from 3–56 days in advance of the experiment to determine humoral immunity to A/WI before undergoing further screening against inclusion and exclusion criteria (S4 Text). Volunteers needed to be healthy, between the ages of 18 and 45 years, not living with anyone deemed at high risk of influenza complications on discharge, and not to have had a seasonal influenza vaccine in the last 3 years. Blood samples from volunteers were collected immediately before quarantine entry for repeat serology, although results were not available until after the study. An initial screening haemagglutination inhibition assay (HAI) titre of ≤10 was considered evidence of susceptibility to infection.

### Power calculation

We calculated, based on the reported SAR of the proof-of-concept study [15] (S4 Text), that over a range of scenarios the statistical power of the whole study would be 63%-84%, typically 80% in the most realistic scenario, if a sample size of 125 Recipients was achieved (70 IR, 55 CR). To increase the SAR from that observed in the proof-of-concept study, we opted to inoculate 4 (rather than 2) Donors per 5 Recipients and to conduct exposure events on days 1–4 (rather than 2–3) post inoculation.

## Pre-exposure

Screened, eligible, volunteers entered a closed, quarantine unit on Day -2 and were randomised to Donor or Recipient (IR and CR) groups; thereafter Donors and Recipients were segregated. Donors and Recipients were immediately screened for a panel of 7 imported 'contaminant' respiratory viruses (influenza A, B; adenovirus; respiratory syncytial virus; parainfluenza 1, 2, 3) by a direct fluorescence antibody assay (DFA) (LIGHT DIAGNOSTICS SimulFluor1 Respiratory Screen, Merck Millipore) and any with a positive test were immediately discharged. On day 0, Donors were inoculated intranasally, via pipette in a supine position, with 0.5ml per nostril of a suspension containing 5.5 $\log_{10}TCID_{50}$/ml of influenza A/WI [16,27].

## Exposure events

The study was conducted in three, separate, identically-designed quarantine events (Q1, Q2, and Q3). From Day 1 to Day 4 of each quarantine event, all volunteers took part in an Exposure Event (EE). Individual Donors and Recipients were each allocated to a single exposure room per day where they interacted at close distances for approximately 15 hours/day, for four consecutive days. In-room staff supervised activities such as playing board games, pool, and table football, and watching films, whilst ensuring that volunteers mixed freely, and that IRs complied with face shield use, hand sanitisation, and no-touch-face rules. Donor, IR and CR groups were moved into different corners of the rooms for meal breaks, and Donor and Recipient groups were housed separately at night, including further separation and withdrawal of any Recipients with symptoms to prevent any contamination of the results by Recipient-Recipient secondary transmission. Five exposure rooms were used ranging from 17-30m$^2$ floor area and 50-87m$^3$ volume. Four Donors were non-randomly allocated to each exposure group to ensure even distribution of subjects actively shedding virus. This was achieved by assessing symptom scores and the results of influenza rapid tests (Quidel Sofia) performed on Days 1 and 2. From Day 2 onwards, Donors remained in their allocated group and were not redistributed further. Once assigned to an exposure group, Recipients remained in the same group until the end of the EE or until they developed ILI and were withdrawn to a separate isolation area. On each day of EE, each exposure group rotated to a different exposure room.

## Environmental controls

Each exposure room was assessed pre-quarantine by building and ventilation engineers and modified to achieve a ventilation rate of approximately 4L/second/person (based on planned occupancy during EEs), temperature 18–22˚C, and relative humidity 45–65%, to produce conditions favourable to influenza transmission [13], balanced against tolerability for occupants, and the capability of building systems to provide controlled environments comparable across all three quarantine studies. During each EE, rooms were monitored at 5-minute intervals for $CO_2$ concentration (as a proxy for ventilation rate), temperature and humidity; heating, cooling and humidity were remotely adjusted to maintain optimal conditions.

## Clinical assessment

All subjects underwent thrice daily monitoring of respiratory and systemic symptoms; each symptom was reported as grade 0 (not present) to 3 (severe). Paired venous blood specimens for serology were taken on Day -2 and Day 28. NPS were taken daily from all subjects. Respiratory specimens were analysed by quantitative reverse-transcriptase PCR (qRT-PCR) and serological specimens by HAI and microneutralisation (MN) assays. The qRT-PCR and HAI were

performed in duplicate at the MRC University of Glasgow Centre for Virus Research (with Fast Track Diagnostics qRT-PCR kit) and the U.S. Centers for Disease Control and Prevention (CDC), Atlanta; MN assays were performed by CDC as described previously [28].

### Exhaled breath sampling

Donors were assigned to provide exhaled breath samples on two, randomly selected days within the exposure event, collected using a Gesundheit-II cone collection apparatus allowing for fractionation of particle sizes into 'fine' <5μm and 'coarse' ≥5μm aerosol samples [8,29]. Each sample collection session lasted for 30 minutes. Breath samples were concentrated, extracted, stored at -80˚C, and evaluated by qRT-PCR using protocols and materials specified by the CDC with RNA copy number estimated as previously described [7].

### Discharge

After completion of the EE, Donors were discharged on active treatment with oseltamivir (75mg b.i.d. 5 days), whereas Recipients were observed for 7 days, then discharged with oseltamivir on day 8. All volunteers attended for 28-day (+/-3 days) post virus exposure outpatient follow-up and study dismissal.

### Outcome definitions

Respiratory symptoms were defined as self-reported grade ≥1 of runny nose, stuffy nose, sneeze, sore throat, cough, or shortness of breath 'lasting ≥24 hours' (S4 Text). Fever was defined as temperature >37.9 °C. Symptomatic was defined as evidence of any respiratory symptom lasting ≥24 hours during study days 1–6. Influenza-like Illness (ILI) was defined as an illness >24 hours duration with: either fever and at least one respiratory symptom; or two or more symptoms of grade ≥1, one of which must have been respiratory; eligible non-respiratory symptoms were headache, muscle/joint ache, and malaise; lasting ≥24 hours means report of a respiratory symptom during 3/3 observations within a single day, or at least once per day over two consecutive days. Laboratory confirmed infection was defined as: a 4-fold or greater rise in HAI or MN titres between Day -2 (baseline) and Day 28; or two or more positive NPS test results by qRT-PCR. These differed from the proof-of-concept study, which used seroconversion or a single positive nasal wash (S6 Text).

### Statistics

Comparisons were made between groups using Fisher's exact tests for binomial proportions and between observed and expected outcomes using binomial tests. Tests were two-tailed. All data to reproduce analyses is available from the EMIT Consortium upon request.

## Supporting information

**S1 Text. EMIT Consortium team members.**
(DOCX)

**S2 Text. Role of funding source.**
(DOCX)

**S3 Text. Efficacy of a face shield to reduce transmission of influenza virus in large droplets.**
Contains Figs A-E.
(DOCX)

**S4 Text. Full inclusion and exclusion criteria, power calculation, and sample handling.**
(DOCX)

**S5 Text. Baseline characteristics.** Contains Table A.
(DOCX)

**S6 Text. Comparing the current study and proof-of-concept study using current outcome criteria and applying infection criteria from the proof of concept study.** Contains Tables B-D.
(DOCX)

## Acknowledgments

The authors thank Blanca Beato Arribas and William Booth of BSRIA Limited, Chang Chen, Sheryl Ehrman, Neil Ferguson, Lisa Grohskopf, F. Liaini Gross, Andrew Hayward, Ashley Kang, Jacqueline Katz, John Oxford, Massimo Palmarini, Walt Adamson, Jennifer Wang, and the hVivo clinical team for their advice and contributions to the study. The authors are grateful to other Scientific Advisory Board members: Allan Bennett, Ben Cowling, and Arnold Monto.

## Author Contributions

**Conceptualization:** Jonathan S. Nguyen-Van-Tam, Ben Killingley, Joanne Enstone, Donald K. Milton.

**Data curation:** Jonathan S. Nguyen-Van-Tam, Ben Killingley, Joanne Enstone, Michael Hewitt, Jovan Pantelic, Michael L. Grantham, P. Jacob Bueno de Mesquita, Robert Lambkin-Williams, Anthony Gilbert, Alexander Mann, John Forni, Catherine J. Noakes, Min Z. Levine, LaShondra Berman, Stephen Lindstrom, Simon Cauchemez, Werner Bischoff, Raymond Tellier, Donald K. Milton.

**Formal analysis:** Jonathan S. Nguyen-Van-Tam, Ben Killingley, Joanne Enstone, Michael Hewitt, Michael L. Grantham, P. Jacob Bueno de Mesquita, Alexander Mann, Catherine J. Noakes.

**Investigation:** Jonathan S. Nguyen-Van-Tam, Ben Killingley, Joanne Enstone, Michael Hewitt, Jovan Pantelic, Michael L. Grantham, P. Jacob Bueno de Mesquita, Robert Lambkin-Williams, Anthony Gilbert, Alexander Mann, John Forni, Catherine J. Noakes, Min Z. Levine, LaShondra Berman, Stephen Lindstrom, Simon Cauchemez, Werner Bischoff, Donald K. Milton.

**Methodology:** Jonathan S. Nguyen-Van-Tam, Ben Killingley, Joanne Enstone, Michael Hewitt, Jovan Pantelic, Michael L. Grantham, P. Jacob Bueno de Mesquita, Robert Lambkin-Williams, Anthony Gilbert, Alexander Mann, John Forni, Catherine J. Noakes, Min Z. Levine, LaShondra Berman, Stephen Lindstrom, Simon Cauchemez, Werner Bischoff, Raymond Tellier, Donald K. Milton.

**Writing – original draft:** Jonathan S. Nguyen-Van-Tam, Ben Killingley, Joanne Enstone, P. Jacob Bueno de Mesquita, Robert Lambkin-Williams, Alexander Mann, Donald K. Milton.

**Writing – review & editing:** Jonathan S. Nguyen-Van-Tam, Ben Killingley, Joanne Enstone, Michael Hewitt, Jovan Pantelic, Michael L. Grantham, P. Jacob Bueno de Mesquita, Robert Lambkin-Williams, Anthony Gilbert, Alexander Mann, John Forni, Catherine J. Noakes, Min Z. Levine, LaShondra Berman, Stephen Lindstrom, Simon Cauchemez, Werner Bischoff, Raymond Tellier, Donald K. Milton.

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
