## [Decision Letter · Decision Letter 0]

11 Apr 2020

Dear Dr. Killingley,

Thank you very much for submitting your manuscript "Minimal transmission in an influenza A (H3N2) human challenge-transmission model within a controlled exposure environment" for consideration at PLOS Pathogens. As with all papers reviewed by the journal, your manuscript was reviewed by members of the editorial board and by several independent reviewers. In light of the reviews (below this email), we would like to invite the resubmission of a significantly-revised version that takes into account the reviewers' comments.

We cannot make any decision about publication until we have seen the revised manuscript and your response to the reviewers' comments. Your revised manuscript is also likely to be sent to reviewers for further evaluation.

Sincerely,

Peter Palese

Associate Editor

PLOS Pathogens

Ron Fouchier

Section Editor

PLOS Pathogens

Kasturi Haldar

Editor-in-Chief

PLOS Pathogens

orcid.org/0000-0001-5065-158X

Michael Malim

Editor-in-Chief

PLOS Pathogens

orcid.org/0000-0002-7699-2064

Reviewer's Responses to Questions

**Part I - Summary**

Reviewer #1: This manuscript summarizes the findings of three separate human influenza virus transmission studies performed in a highly controlled setting. While the authors found consistent shedding of the H3N2 challenge virus in nasal swab samples in almost all directly challenged individuals up to day 4, only 1 out of 75 recipients (including protected and unprotected individuals) was confirmed to be infected. Interestingly, the study found substantially lower rates of aerosolized particle shedding in directly infected individuals compared to individuals in a previous study in a natural infection setting. The study could be considered largely a "failed" experiment, but is nonetheless important since it raises important questions on how controlled human challenge studies may substantially differ from natural infection. The viral titers required to successfully infect individuals in challenge studies are generally higher than what is thought to be required in natural infection and based on this study the experimental infection may be even more different from natural infection than previously thought.

Reviewer #2: Overall this manuscript describes the result of a human challenge transmission study that did not go as planned. While individuals who were inoculated with virus became infected, there was minimal transmission to contacts. While the study essentially describes negative results, there is great interest in these data with such studies being rarely conducted. Having said this, the authors main conclusion that ventilation was a key factor is not convincingly supported and other alternatives need further exploration.

1. Ln 154. A bit more detail on exactly what constitutes “seronegative (HAI≤10) volunteers” would be informative. Were they seronegative to just the challenge antigen, a range of viruses of the same subtype, a range of viruses of different subtypes etc.

2. Ln 158. EE should be defined here (first mention in text).

3. Ln 173. The authors describe that infection produced an infection rate of 81% 174 (42/52) among inoculated volunteers. Is this typical of H3N2 experimental infections? Data form other studies in presented in ln 285, but a larger comparison to the literature would be of interest.

4. Ln 181. I am a bit confused by the discussion of the ten Donors that had greater than anticipated immunity on admission. What exactly does this mean? Does it mean that they were infected naturally between having admission blood drawn and actually entering quarantine? What viruses were they confirmed positive with?

5. Ln 216. What were the Ct valus of the two contacts that had transient PCR positivity?

6. A quarter of the CR recipients showed signs of ILI? What was the reason for this? Was testing done for any other organism? Could this possibly explain the very low rates of virus transmission, i.e., non-specific antiviral responses due to infection with other respiratory pathogens?

7. The authors state a number of variable that could have impacted the lower than expected transmission rates in the study but tend to favor ventilation as the likely factor. I am a little less than convinced by their arguments and am more inclined to question the high levels of ILI and symptoms in recipients. This finding must be further discussed in the context of the overall findings.

**Part II – Major Issues: Key Experiments Required for Acceptance**

Reviewer #1: - A major concern is that serology was only assessed based on HAI and neutralization. However, recent publications have shown that antibodies measured by ELISA can not only be predictive of protection, but may also be a more sensitive tool to assess seroconversion in the absence of HAI titer increase (PMID 31160818). It would substantially improve the manuscript if the authors could measure antibody responses against HA and NA in pre- and post-challenge samples to identify potential pre-existing titers or seroconversion that could not be detected in the traditional assays.

- It would be helpful if the authors could comment in more detail on the aerosol shedding in naturally infected individuals. Is it possible that some selection bias might result in more severe cases to be enrolled, who might spread more aerosolized virus due to lower respiratory tract infection?

Reviewer #2: (No Response)

**Part III – Minor Issues: Editorial and Data Presentation Modifications**

Reviewer #1: The title could be changed to more explicitly state intranasal droplet delivery as the route of administration, which may be an important factor in the limited viral spread via droplets.

Reviewer #2: (No Response)

PLOS authors have the option to publish the peer review history of their article (what does this mean?). If published, this will include your full peer review and any attached files.

Reviewer #1: No

Reviewer #2: No
---

## [Editor Report · Decision Letter 1]

14 Jun 2020

Dear Dr. Killingley,

We are pleased to inform you that your manuscript 'Minimal transmission in an influenza A (H3N2) human challenge-transmission model within a controlled exposure environment' has been provisionally accepted for publication in PLOS Pathogens.

Best regards,

Peter Palese

Associate Editor

PLOS Pathogens

Ron Fouchier

Section Editor

PLOS Pathogens

Kasturi Haldar

Editor-in-Chief

PLOS Pathogens

orcid.org/0000-0001-5065-158X

Michael Malim

Editor-in-Chief

PLOS Pathogens

orcid.org/0000-0002-7699-2064
---

## [Editor Report · Acceptance letter]

9 Jul 2020

Dear Dr. Killingley,

We are delighted to inform you that your manuscript, "Minimal transmission in an influenza A (H3N2) human challenge-transmission model within a controlled exposure environment," has been formally accepted for publication in PLOS Pathogens.

Best regards,

Kasturi Haldar

Editor-in-Chief

PLOS Pathogens

orcid.org/0000-0001-5065-158X

Michael Malim

Editor-in-Chief

PLOS Pathogens

orcid.org/0000-0002-7699-2064